# Predictors of COVID-19 vaccination intention among students in Ghana: An application of the Health Belief Model and Theory of Planned Behaviour

Hudatu Ahmed[1], Mawuli Gohoho[2,3]*, Samuel Adolf Bosoka[3,4], George Sarpong Agyemang[3,5], Sorengmen Amos Ziema[3,6], James Alorwu[3,7], Isaac Annobil[2], Veronica Okwuchi Charles-Unadike[1]

**1** Fred N. Binka School of Public Health, University of Health and Allied Sciences, Hohoe, Volta Region, Ghana, **2** Jasikan Municipal Health Directorate, Ghana Health Service, Jasikan, Oti Region, Ghana, **3** Ghana Field Epidemiology and Laboratory Training Programme, Accra, Ghana, **4** Disease Surveillance Unit, Volta Regional Health Directorate, Ghana Health Service, Ho, Volta Region, Ghana, **5** Department of Public Health, Catholic University of Ghana, Fiapre, Ghana, **6** Department of Health Information and Records Management, Ho Teaching Hospital, Ministry of Health, Ho, Volta Region, Ghana, **7** Tumu Municipal Hospital, Ghana Health Service, Tumu, Upper West Region, Ghana

* mawuli.gohoho@ghs.gov.gh

## Abstract

Students are an important group in COVID-19 prevention, as their vaccination decisions influence peers and families and contributes to broader community protection. However, limited evidence exists on factors influencing their decisions to get vaccinated. This study examined COVID-19 vaccination intention and its associated predictors among students in Jasikan Municipality, Ghana, using the Health Belief Model (HBM) and Theory of Planned Behaviour (TPB) to guide strategies for improving vaccine acceptance. A cross-sectional study was conducted between 25 July and 5 August 2022 using a multistage sampling technique to select participants from Senior High Schools. Using a pilot-tested questionnaire, data were collected on socio-demographics, COVID-19 protocol adherence, health-related characteristics, HBM and TPB constructs, and vaccination intention. The constructs were measured using Likert-scale items adapted from previous studies. Reliability was assessed using Cronbach's alpha ($\alpha = 0.63$-$0.86$). Descriptive and inferential statistics were performed. Hierarchical binary logistic regression was used to assess the predictors of COVID-19 vaccination intention with statistical significance set at $P < 0.05$. We found that 206 (49.0%) of the 420 students interviewed were females, and 141 (33.6%) were aged 15–17 years. The mean age was 18.21 (±1.79) years. Overall, 257 (61.0%) students expressed intention to vaccinate against COVID-19. Previous COVID-19 experience (AOR = 3.26; 95%CI: 1.11-9.56), adherence to social distancing (AOR = 1.85; 95% CI: 1.17-2.91), attitude towards vaccination (AOR = 1.10; 95%CI: 1.03-1.18), subjective norms (AOR = 1.09; 95%CI: 1.04-1.15),

**Data availability statement:** All data analysed during this study are included in this article and its Supporting information files. The anonymised dataset is provided as Supporting information (S1 Data).

**Funding:** The author(s) received no specific funding for this work.

**Competing interests:** The authors have declared that no competing interests exist.

and self-efficacy (AOR = 1.20; 95%CI: 1.11-1.31) were significant predictors of COVID-19 vaccination intention. These findings indicate that six in ten students were willing to receive the COVID-19 vaccine with intention influenced by prior COVID-19 experience, social distancing behaviour, and key constructs from HBM and TPB. School-based vaccination programs should prioritize attitude change interventions that leverage peer influence and teacher support while building students' confidence toward vaccination to improve vaccine uptake.

## Introduction

The WHO declared COVID-19 a global public health emergency on 30 January 2020 [1]. This affected over 770 million people globally, causing more than 7 million deaths and disruptions to healthcare systems and economies [2]. Although vaccinations are a major public health advancement preventing millions of deaths each year including those from COVID-19, their impact in controlling pandemics depends on public acceptance [3,4]. The delay of the public in acceptance or refusal despite the availability of vaccination services has emerged as a major global health concern even before the COVID-19 pandemic [5,6]. For instance, polio vaccinations were boycotted in Nigeria and Kenya due to false claims and misinformation about the vaccine [7–9]. For the COVID-19 vaccine, fear of contracting an infection, concerns about side effects, beliefs about microchips or infertility, underestimation of disease impact, and a general lack of trust in government officials, state institutions, and pharmaceutical companies were reported as reasons for hesitancy [10–13].

Students constitute a vital and distinct demographic group within the population [14–17]. They act as potential "messengers" in disseminating effective health-promoting behaviours to their families and communities [14,18]. The environment within which students find themselves and their social interactions could fuel outbreaks of COVID-19 disease and increase local transmissions [14,19]. For instance, in the Jasikan Municipality, 54.5% of the COVID-19 cases were among students [20]. This suggests a high incidence of COVID-19, which could increase the risk of complications and adverse clinical outcomes, while also negatively impacting academic performance, as reported in previous studies [21–23].

In response to such trends and as part of measures to increase protection among adolescents, the Pfizer-BioNTech COVID-19 vaccine was recommended for use in persons aged 15 years and above in schools to help stop the transmission and increase the level of protection against the virus. The Ghana Health Service and the Food and Drugs Authority granted approval for use of the vaccine to cover children from 15 years and above. This recommendation was communicated in a letter dated 19 November 2021, from the Director-General of the Ghana Technical and Vocational Education Training (TVET) Service, addressed to all Heads of TVET institutions (Public and Private), requesting them to facilitate the vaccination of all learners aged 15 or older in their institutions across the country.

However, the current study was conceptualised prior to the release of this national directive to address the significant burden of COVID-19 among students and to gather information on their vaccination intentions for targeted health promotion interventions. In Jasikan, limited evidence exists on factors influencing students' decisions regarding the uptake of COVID-19 vaccines. To comprehensively assess these intentions, the study integrated the Health Belief Model (HBM) and the Theory of Planned Behaviour (TPB) (Fig 1).

HBM explains preventive behaviours through individual beliefs, including perceived susceptibility, perceived severity, perceived benefits, perceived barriers, cues to action, and self-efficacy and highlights how students evaluate health threats and the value of vaccination [24–26]. TPB, on the other hand, focuses on behavioural intention as the most direct predictor of action, incorporating attitudes, subjective norms, and perceived behavioural control to capture both motivation and perceived ability to act [27,28]. Evidence from recent studies indicates that integrating both the HBM and the TPB provides strong predictive power for COVID-19 vaccination intention [29,30]. The study therefore assessed the COVID-19 vaccination intention of Senior High School students in Jasikan Municipality of Ghana, examined how

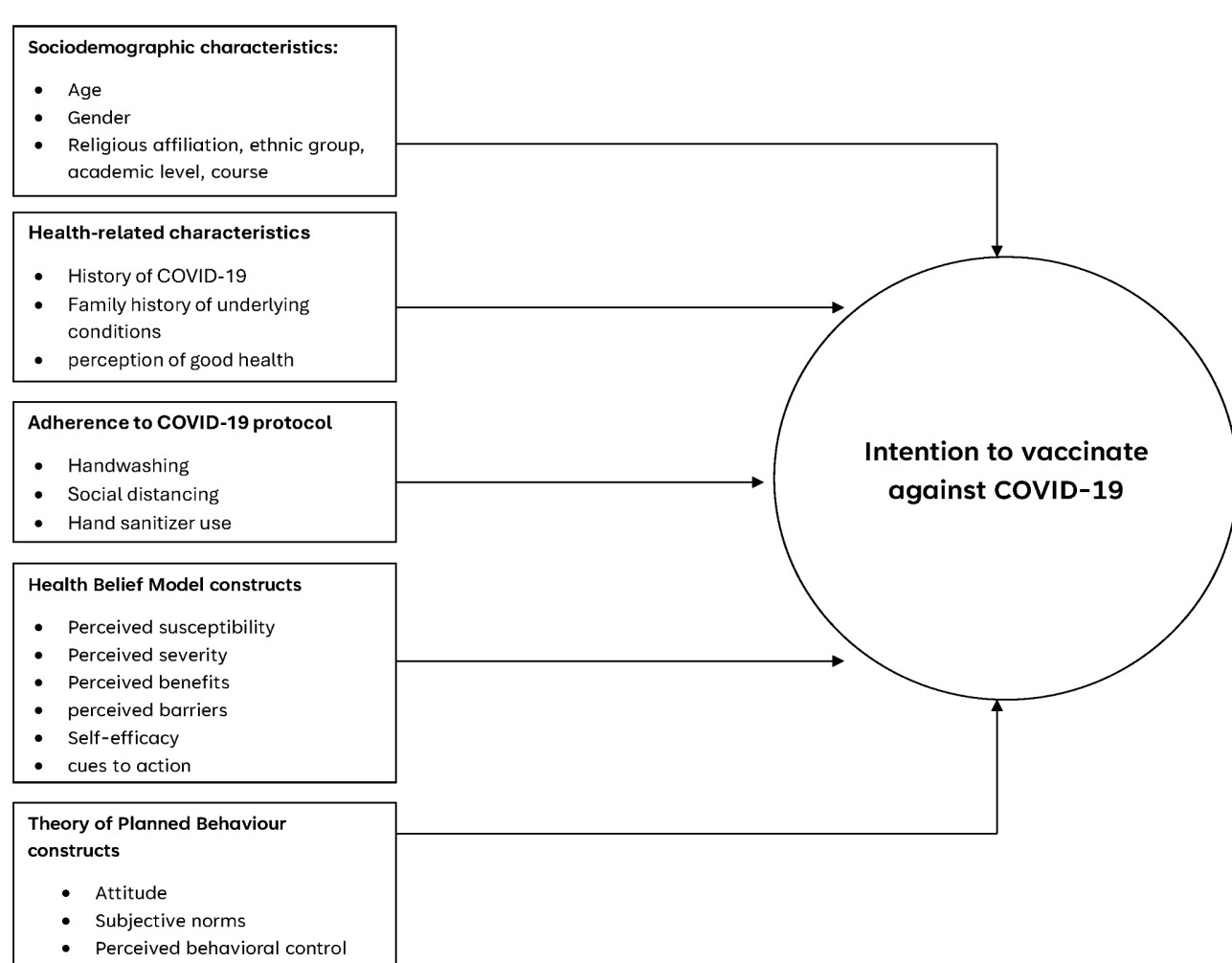

**Fig 1. Conceptual framework showing how HBM and TPB constructs, along with socio-demographic, health-related, and COVID-19 protocol adherence factors, influence students' intention to vaccinate against COVID-19.**

socio-demographics, health-related characteristics, adherence to COVID-19 protocols, and constructs from the HBM and the TPB predict this intention.

## Methods

### Ethics statement

Ethical approval was obtained from the University of Health and Allied Sciences Research Ethics Committee (UHAS-REC A.10 [23] 21–22). Permission was granted by the Jasikan District Office, Ghana Education Service, to access the schools for data collection. Participation was voluntary, and participants could withdraw at any time without consequence. For those aged 15–17 years, parental or guardian consent was secured in addition to written assent from the participants themselves. For those aged 18 years and above, written informed consent was obtained. The study purpose, risks, and benefits were clearly explained to all participants. There were no known risks or direct benefits. No compensation was provided, though participants were appreciated for their time. Data were anonymised, stored securely with password protection on a personal laptop accessible only to the principal investigator.

### Study site

The study was conducted in four SHSs: Bueman SHS, Okadjakrom Senior High Technical School (SHTS), Father Dogli SHS, and Baglo Ridge SHS, all located in Jasikan Municipality in the Oti Region of Ghana. The municipality shares boundaries with the Kadjebi District to the north, the Biakoye District to the west, the Guan District to the south, and the Republic of Togo to the east (Fig 2).

### Study design and population

A cross-sectional study was conducted among SHS students in Jasikan Municipality, grounded in the HBM and TPB as applied in previous studies [29,30]. This design was chosen because of its ability to quickly assess vaccination intention and multiple associated factors at a single point in time, though it does not allow for causal inference. The study population included students aged 15 years and above who were present during the data collection period. The inclusion of students aged 15 years and above was based on the national guideline recommending the use of COVID-19 vaccine among persons in this age group in schools in Ghana. However, those who were severely ill, absent, or had mental disabilities were excluded from the study. This was to ensure active participation with informed responses without compromising their wellbeing or data quality.

### Sample size and sampling method

The sample size was computed using the single population proportion formula [31], $n = \frac{(z_{\alpha/2})^2 \times p(1-p)}{d^2}$ with the following assumptions: an estimated proportion of persons aged 15 years and above with the intention to vaccinate against COVID-19 in Ghana (p = 51%) [32]; Confidence level at 95% (Zα/2) = 1.96, a margin of error (d) = 0.05 and non-response rate = 10%. The minimum sample size determined was 420. To ensure representativeness and comparability, a multi-stage sampling procedure was used in the selection of study participants. At stage one, all the four SHSs were involved. At stage two, the students from the schools were stratified into SHS 2 and SHS 3 to have 8 strata from the four schools. During the period of data collection, only those in SHS 2 and SHS 3 were available. In the third stage, students were selected from each stratum proportional to its population size. In the fourth stage, simple random sampling was applied within each stratum to select eligible participants to have 420 students. All selected participants (100%) responded to the survey and therefore, no additional handling for non-response was required.

### Data collection procedure

A pretested questionnaire was used to collect data from participants from 25 July to 5 August 2022. Amidst the COVID-19 pandemic, trained research assistants were educated on prevention and provided with face masks and hand

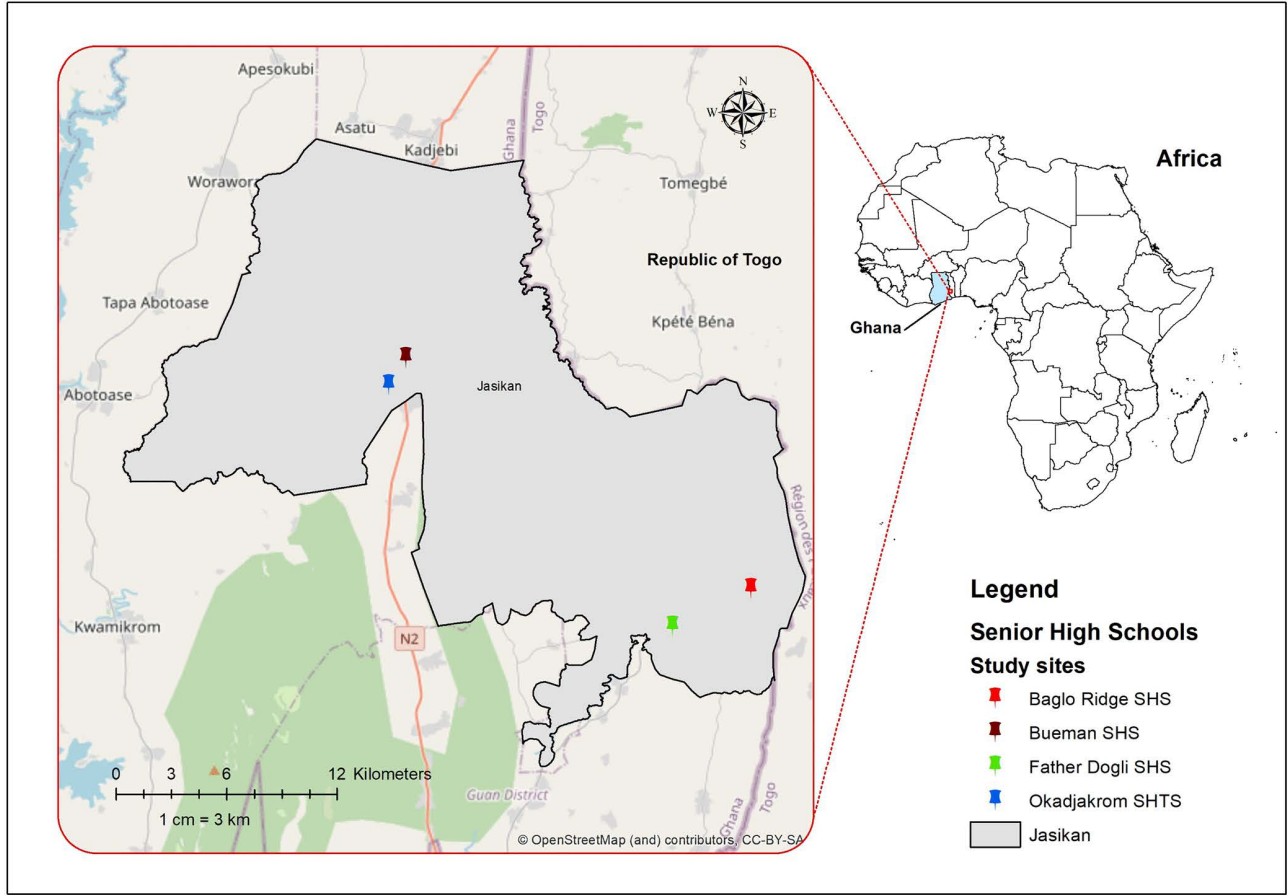

**Fig 2. Map of the study area showing senior high schools where data was collected in Jasikan Municipality, Ghana.** (The base layer shapefile was obtained from GADM (https://gadm.org/download_country.html) and visualized using ArcMap 10.4. GADM permits academic use, including publication of maps in open-access journals under CC-BY licenses. License terms available at https://gadm.org/license.html).

sanitisers to reduce the probability of person-to-person transmission. Having explained the purpose of the study to the participants and obtaining consent, research assistants distributed the questionnaires in the schools visited. However, as the data were self-reported and completed directly by participants, the research assistants were not involved in recording responses and were not blinded to the study hypotheses. As all participants could speak and understand English, no translation was required. The self-administered questionnaire was pretested among a small group of students outside the study sample and adjustments were made for clarity and comprehension. Adapted from previous studies [29,33,34], the questionnaire included sections on sociodemographic characteristics, health-related variables, COVID-19 protocol adherence, HBM, TPB and COVID-19 vaccination intention. The reliability of the items used to measure the HBM and TPB constructs was assessed using Cronbach's alpha ranged from 0.63 to 0.86 which was based on the dataset obtained for the study (Table 1).

## Variables and measurements

The dependent variable was COVID-19 vaccination intention, measured with a single 5-point Likert item (1 = Certain, 2 = Very Likely, 3 = Somewhat Likely, 4 = Not Likely, 5 = Never). This was recoded into a binary outcome (1 = Intends to get

**Table 1. Reliability for HBM and TPB constructs related to COVID-19 vaccination intention.**

| Measures | Items | Mean | std | Cronbach's Alpha (α) |
|---|---|---|---|---|
| **Model 1: Health Belief Model** | | | | |
| Perceived Severity | If I should get infected with COVID-19…...<br>• I do not think it will cause too much suffering<br>• the chance of recovering from it is very high<br>• the chance of dying is very high<br>• I can be hospitalised for weeks | 13.73 | 2.52 | 0.66 |
| Perceived Susceptibility | I believe that if I do not get vaccinated the chance of…<br>• getting COVID-19 currently is high<br>• getting COVID-19 in future is high<br>• my family getting COVID-19 is high<br>• my friends getting COVID-19 is high | 11.72 | 4.36 | 0.82 |
| Perceived Benefit | I believe that getting vaccinated against COVID-19…<br>• will decrease my chances of getting COVID-19<br>• will improve my health and improve my academics<br>• will prevent suffering and complications of COVID-19 | 10.41 | 3.18 | 0.74 |
| Perceived Barrier | • Going to get vaccinated is expensive<br>• Getting vaccinated requires time and effort<br>• The COVID-19 vaccine can cause some reaction to my body<br>• I do not have a national ID for the vaccination<br>• The COVID-19 vaccine can cause infertility<br>• The COVID-19 vaccine can kill | 13.24 | 4.41 | 0.75 |
| Self-Efficacy | • If I take all the necessary precautions (handwashing, face mask usage etc.), I do not need to be vaccinated against COVID-19<br>• I can receive the COVID-19 vaccine on time<br>• It is not difficult for me to receive the COVIID-19 vaccine | 9.92 | 2.94 | 0.63 |
| Cues to Action | • The chances of me getting vaccinated against COVID-19 will increase if…<br>• celebrities and famous people on social media express support for the benefit of the vaccine<br>• friends and family express support for the benefit of the vaccine<br>• my teachers and headteachers advise me to take it<br>• it does not cause serious problems to vaccinated people<br>• I get complete and credible<br>• information on the vaccine | 16.46 | 4.65 | 0.79 |
| **Model 2: Theory of planned behavior** | | | | |
| Attitude | • Getting vaccinated is a tedious process that requires time and effort<br>• COVID-19 vaccine is beneficial for the students<br>• COVID-19 vaccine is effective in preventing the disease<br>• Once a COVID-19 vaccine is recommended, getting it would be good<br>• COVID-19 vaccine is the mark of the beast (666)<br>• People will die from taking the COVID-19 vaccine after 2 years | 19.41 | 3.69 | 0.69 |
| Subjective Norms | • Most of my friends will take the COVID-19 vaccine<br>• My family who are important to me would approve of me getting any available COVID-19 vaccine<br>• My relatives who are important to me would approve of me getting any available COVID-19 vaccine<br>• My friends who are important to me would approve of me getting any available COVID-19 vaccine<br>• My headmaster/teachers who are important to me would approve of me getting any available COVID-19 vaccine | 16.51 | 4.97 | 0.86 |
| Perceived Behavioural Control | • I could easily receive a COVID-19 vaccine if I wanted to<br>• It is mostly up to me whether or not I have COVID-19 vaccination | 7.36 | 2.09 | 0.69 |

vaccinated; 0 = Does not intend to get vaccinated), with responses 1–3 coded as 1 and 4–5 coded as 0. Though dichotomisation may lead to some loss of information, it provides clearer interpretation and enables straightforward comparison of factors associated with vaccination intention, as applied in earlier studies [29,33,34].

The independent variables were organized into five blocks:

- **Socio-demographic:** included (1) age (categorised into 15–17 and 18–30 years from an original numeric variable to examine age group differences), (2) gender, (3) religious affiliation, (4) ethnic group, (5) school attending, (6) academic level, (7) residency status and (8) course of study (recategorised into a new variable with three groups: 1 = academic [reference group: business, general arts, general science], 2 = technical/applied [technical, general agriculture, home economics], 3 = creative/vocational [fashion design, visual art]) (Fig 1).

- **Health-related:** comprised (1) history of COVID-19, (2) family history of underlying disease conditions, and (3) perception of good health (Fig 1).

- **COVID-19 protocol adherence:** included (1) face mask usage, (2) adherence to handwashing, (3) adherence to social distancing practices and (4) hand sanitiser usage (Fig 1).

- **HBM constructs** (Table 1): consisted of (1) perceived susceptibility (four items, Cronbach's α = 0.82), (2) perceived severity (four items, Cronbach's α = 0.66), (3) perceived benefits (three items, Cronbach's α = 0.74), (4) perceived barriers (six items, Cronbach's α = 0.75), (5) self-efficacy (three items, Cronbach's α = 0.63) and (6) cues to action (five items, Cronbach's α = 0.79).

- **TPB constructs** (Table 1): included (1) attitude (six items, Cronbach's α = 0.69), (2) subjective norms (five items, Cronbach's α = 0.86), (3) and perceived behavioural control (PBC) (two items, Cronbach's α = 0.69).

The items used to measure the HBM and TPB constructs were adapted from previously validated instruments used in similar studies on vaccine acceptance and health behaviour [29,33,34]. Minor modifications were made to tailor the items to the COVID-19 vaccination context and the target population for the study. The items were measured on a 5-point Likert scale ranging from 1 (strongly disagree) to 5 (strongly agree). Negatively worded items were reverse-scored. Mean scores were computed for each item to derive the independent HBM and TPB constructs (Table 1).

## Data management and analysis

Data were checked for completeness and consistency before entry into EpiData version 3.1 and exported to STATA version 16 (StataCorp LLC, College Station, Texas, USA) for analysis. Any field that appeared blank was cross-checked with the hardcopy questionnaire and completed to address missingness. Descriptive analysis involved computation of means, standard deviations, frequencies, and proportions to summarise key variables. Chi-square tests assessed associations between categorical variables and COVID-19 vaccination intention, while independent sample t-tests compared mean scores of HBM and TPB constructs between students based on vaccination intention. Effect sizes were estimated using Cohen's d and interpreted according to Cohen's benchmarks [35], where 0.20 indicates a small effect, 0.50 a moderate effect, and 0.80 a large effect. Negative values of Cohen's d reflected the direction of group differences, while magnitudes were interpreted using absolute values.

Hierarchical binary logistic regression analysis was conducted in three steps to identify predictors of vaccination intention. First, univariate analyses were performed for all independent variables to assess their individual association with vaccination intention. Variables with $p < 0.05$ in univariate analyses were included in the hierarchical binary logistic regression models. Model 1 included sociodemographic, health-related, adherence, and HBM variables. Model 2 included the same variables, replacing HBM variables with TPB constructs. Model 3 combined all variables from the previous models. Sociodemographic variables were adjusted for in all models. HBM and TPB constructs were treated as continuous variables. Adjusted odds ratios (AOR) with 95% confidence intervals were reported. Multicollinearity was monitored throughout the hierarchical modelling process using variance inflation factors (VIF). VIF values were computed for variables at each step by regressing each predictor on all other predictors included up to that step. Mean VIF values were 1.15 (Model 1), 1.11

(Model 2), and 1.20 (Model 3), with all individual VIF values remaining below 5 confirming no problematic multicollinearity at any stage of model building. Goodness of fit was assessed using log-likelihood values, LR chi-square tests with related p-values, and pseudo $R^2$ measures including Cragg Uhler's value (S2 Table).

## Results

### Students' characteristics

A total of 420 SHS students participated in the study with a response rate of 100%. Of these, 206 (49.0%) were females and 214 (51.0%) were males. Most of the students were aged 18–30 years [279 (66.4%)], while 141 (33.6%) were aged 15–17 years. The mean age was 18.21 (±1.79) years. Most participants were in SHS 2 [248 (59.0%)], followed by SHS 3 [172 (41.0%)]. The academic programmes pursued by participants included General Arts [148 (35.2%)], Technical [135 (32.1%)], Home Economics [60 (14.3%)], General Science [54 (12.9%)], and Business [23 (5.5%)]. Three hundred and sixty-four (86.7%) students were resident within their respective schools, while 56 (13.3%) were non-resident (S1 Table).

### Health-related characteristics

Only 26 (6.2%) of the participants reported ever having COVID-19, while 394 (93.8%) had not. A family history of underlying conditions was reported by 41 (9.8%) participants. Most students [308 (73.3%)] perceived their health status as good (S1 Table).

### Adherence to COVID-19 protocols

Mask usage was reported by 275 (65.5%) of the participants. Three hundred and sixty-nine of the participants (87.9%) practiced regular handwashing, 333 (79.3%) reported hand sanitiser use, and 275 (65.5%) practiced social distancing (S1 Table).

### Intention to vaccinate against COVID-19

Overall, 257 [61.2%; 95% CI (56.4% - 65.8%)] expressed intention to receive the COVID-19 vaccine, while 163 [38.8%; 95%CI (34.2% - 43.6%)] did not (Fig 3).

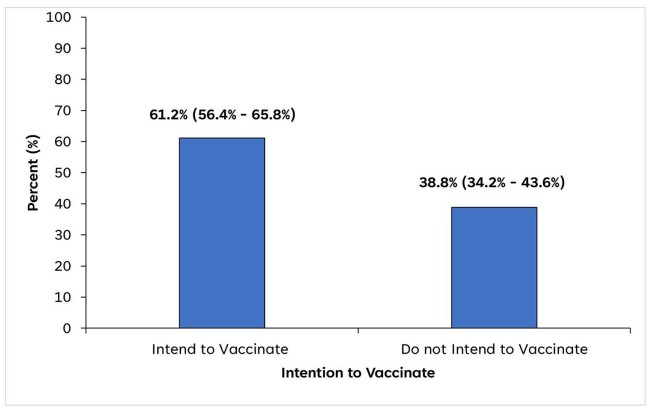

**Fig 3. Percentage of students who intend versus do not intend to vaccinate against COVID-19 in Jasikan Municipality, Ghana, July to August 2022.** The bar chart shows the percentage of students who intend or do not intend to vaccinate against COVID-19. Numbers above bars indicate the proportion (%) with 95% confidence intervals (in parentheses).

## Univariate analysis

For the univariate analysis between sociodemographic variables and intention to vaccinate against COVID-19, age group was significantly associated with intention to vaccinate ($\chi^2 = 3.97$, p = 0.046). Academic programme was also significantly associated with vaccination intention ($\chi^2 = 18.01$, p = 0.012). Among the health-related variables, history of COVID-19 showed a statistically significant association ($\chi^2 = 7.25$, p = 0.007). Handwashing practice ($\chi^2 = 5.40$, p = 0.02) and social distancing ($\chi^2 = 10.15$, p = 0.001) were significantly associated with intention to vaccinate (S1 Table).

For the HBM variables, participants who intended to vaccinate had significantly higher mean perceived susceptibility with a small negative effect (t = -1.99; Cohen's d = -0.20). Perceived benefit was also significantly higher among those intending to vaccinate with a small negative effect (t = -3.60; d = -0.36). Further, the mean self-efficacy was significantly higher among those intending to vaccinate with moderate negative effect (t = -5.05; d = -0.51). No significant differences were observed for mean perceived severity, barriers, or cues to action. Among the TPB constructs, the mean attitude score was significantly higher among those with intention to vaccinate with moderate negative effect (t = -5.47; d = -0.55). Similarly, those with intention to vaccinate reported significantly higher mean subjective norms with moderate negative effect (t = -5.35; d = -0.54). No significant difference was found in perceived behavioural control between the two groups (Table 2).

## Factors associated with intention to vaccinate against COVID-19

Hierarchical binary logistic regression was conducted in three steps to examine the contributions of different sets of predictors. In Model 1, students with history of COVID-19 were 3.55 times more likely to intend to vaccinate than those who had not experienced the infection (AOR = 3.55; 95% CI: 1.25–10.12). Those adhering to social distancing were 1.94 times more likely to intend to vaccinate (AOR = 1.94; 95% CI: 1.25–3.02). Also, each unit increase in perceived benefit increased intention to vaccinate by 11% (AOR = 1.11; 95% CI: 1.03–1.20), and each unit increase in self-efficacy was associated with a 21% higher likelihood of intention to vaccinate (AOR = 1.21; 95% CI: 1.12–1.30) (Table 3). The model explained 19% of the variance in vaccination intention (Cragg & Uhler's $R^2 = 0.19$) (S2 Table).

In Model 2, history of COVID-19 infection (AOR = 2.91; 95%CI: 1.03-8.21) and adherence to social distancing (AOR = 1.73; 95% CI: 1.11-2.69;) remained significant. In addition, students in technical or applied courses were 1.71 times more likely to intend to vaccinate (AOR = 1.71; 95% CI: 1.07–2.73). Each unit increase in attitude increased

**Table 2. Univariate analyses between HBM and TPB constructs and COVID-19 vaccination intention.**

| Variables | COVID-19 vaccination intention | | t-test | p-value (two-tail) | Effect size |
|---|---|---|---|---|---|
| | No (n = 163) | Yes (n = 257) | | | (Cohen's d) |
| **HBM** | Mean (std. dev.) | Mean (std. dev.) | | | |
| Perceived susceptibility | 11.19 (4.30) | 12.06 (4.37) | -1.99 | **0.047*** | -0.20 |
| Perceived severity | 13.80 (2.58) | 13.68 (2.49) | 0.46 | 0.645 | 0.05 |
| Perceived benefit | 9.72 (3.30) | 10.85 (3.02) | -3.60 | **0.0004*** | -0.36 |
| Perceived barrier | 13.75 (4.41) | 12.91 (4.39) | 1.90 | 0.058 | 0.19 |
| Self-efficacy | 9.04 (2.63) | 10.49 (3.00) | -5.05 | **<0.0001*** | -0.51 |
| Cues to action | 16.11 (4.58) | 16.69 (4.71) | -1.25 | 0.212 | -0.13 |
| **TPB** | | | | | |
| Attitudes | 18.21 (3.68) | 20.17 (3.51) | -5.47 | **<0.0001*** | -0.55 |
| Subjective norms | 14.93 (4.68) | 17.51 (4.91) | -5.35 | **<0.0001*** | -0.54 |
| Perceived behavioural control | 7.42 (2.08) | 7.33 (2.11) | 0.42 | 0.673 | 0.04 |

*p-value less than 0.05; std. dev = standard deviation.

vaccination intention by 13% (AOR = 1.13; 95% CI: 1.06–1.20), and each unit increase in subjective norms increased intention by 8% (AOR = 1.08; 95% CI: 1.03–1.13) (Table 3). The model explained 19.8% of the variance in vaccination intention (Cragg & Uhler's R² = 0.198) (S2 Table).

In the final model, students who reported experiencing COVID-19 were 3.26 times more likely to have the intention to vaccinate (AOR = 3.26; 95% CI: 1.11–9.56). Those who adhered to social distancing were 1.85 times more likely to intend to vaccinate (AOR = 1.85; 95% CI: 1.17–2.91). Moreover, each unit rise in attitude increased intention by 10% (AOR = 1.10; 95% CI: 1.03–1.18), and each unit rise in subjective norms also increased intention by 9% (AOR = 1.09; 95% CI: 1.04–1.15). Each unit rise in self-efficacy increased intention by 20% (AOR = 1.20; 95% CI: 1.11–1.31). Perceived susceptibility, perceived benefit, and course of study remained insignificant (Table 3). The Model explained 25.7% of the variance in COVID-19 vaccination intention (Cragg & Uhler's R² = 0.257) (S2 Table).

## Discussion

This study examined the prevalence and predictors of COVID-19 vaccination intention among SHS students in Jasikan Municipality, Ghana, using constructs from the HBM and TPB. A cross-sectional design was used, and data were analysed with hierarchical binary logistic regression to identify significant predictors of vaccination intention.

Overall, more than three-fifth of the students expressed intention to vaccinate against COVID-19, while nearly two-fifth did not. This indicates a majority intention to vaccinate, but also a substantial proportion of vaccine hesitancy. This intention rate is higher than the 54.3% reported in a national survey of Ghanaians conducted between March and April

Table 3. Hierarchical binary logistic regression models predicting COVID-19 vaccination intention.

| Predictor Variables | Model 1: Health-related, Adherence and HBM | | Model 2: Health-related, adherence and TPB | | Model 3: Health-related, Adherence, HBM and TPB | |
|---|---|---|---|---|---|---|
| | AOR (95% CI) | p-value | AOR (95% CI) | p-value | AOR (95% CI) | p-value |
| **Block 1: Sociodemographic** | | | | | | |
| Sex (Ref: Female) | | | | | | |
| Male | 0.78 (0.50 – 1.22) | 0.275 | 0.72(0.46–1.13) | 0.148 | 0.81(0.51 – 1.29) | 0.371 |
| Course (Ref: Academic) | | | | | | |
| Technical/Applied | 1.45(0.91–2.30) | 0.12 | 1.71(1.07–2.73) | **0.025*** | 1.59(0.98 – 2.57) | 0.061 |
| Creative/Vocational | 1.21(0.52–2.79) | 0.661 | 1.38(0.59–3.18) | 0.456 | 1.12(0.47 – 2.67) | 0.804 |
| **Block 2: Health- related** | | | | | | |
| History of COVID-19 (Ref: No) | | | | | | |
| Yes | 3.55(1.25–10.12) | **0.018*** | 2.91(1.03–8.21) | **0.043*** | 3.26(1.11 – 9.56) | **0.032*** |
| **Block 3: Adherence** | | | | | | |
| Handwashing (Ref: No) | | | | | | |
| Yes | 1.51 (0.69 – 3.30) | 0.299 | 1.70(0.79–3.63) | 0.173 | 1.51 (0.68 – 3.33) | 0.313 |
| Social distancing (Ref: No) | | | | | | |
| Yes | 1.94 (1.25 – 3.02) | **0.003*** | 1.73(1.11–2.69) | **0.015*** | 1.85(1.17 – 2.91) | **0.008*** |
| **Block 4: HBM Constructs** | | | | | | |
| Perceived susceptibility | 1.03 (0.97 – 1.09) | 0.342 | | | 1.01(0.95– 1.06) | 0.915 |
| Perceived benefit | 1.11(1.03–1.20) | **0.006*** | | | 1.04 (0.95–1.13) | 0.384 |
| Self-efficacy | 1.21(1.12–1.30) | **<0.001*** | | | 1.20 (1.11-1.31) | **<0.001*** |
| **Block 5: TPB Constructs** | | | | | | |
| Attitude | | | 1.13(1.06–1.20) | **<0.001*** | 1.10(1.03 – 1.18) | **0.004*** |
| Subjective norms | | | 1.08(1.03–1.13) | **0.001*** | 1.09(1.04 – 1.15) | **0.001*** |

*p-value less than 0.05; AOR = Adjusted odds ratio; CI = Confidence interval.

2021 [36], the 45% observed among university students in Ghana [37] and the 31.4% found in the general population in Ethiopia [38]. Other studies reported similar intention rates among Palestinian dental students in early 2021 (57.8%) [39] and college students in India (63.8%) [18]. However, it is lower than the average global pooled intention rate (67.7%) [40] and other studies among the general population in Israel (80%), [29] households in Nigeria (>85%) [41], Italian university students (94.7%) [42] and Indonesia residents (93%) [26].

A possible explanation could be the local COVID-19 epidemiology among the students [20] which may have influenced their perceptions of risk. In addition, the age group studied might have different social dynamics and risk perceptions compared to older adults or general university students. The substantial minority who do not intend to vaccinate highlights the critical need for targeted public health strategies to identify and address specific reasons for reluctance within this demographic to improve vaccine uptake.

The history of COVID-19 infection as a predictor aligns with a study in Indonesia [26] as well as dental students in Palestine [39]. However, it contrasts with a study in India [43] and Germany [44]. Within the HBM, previous infection may heighten perceived susceptibility and perceived severity by making the threat more concrete [13]. Such experience can act as a cue to action, prompting individuals to view vaccination as a necessary step to prevent recurrence [33]. Public health campaigns could incorporate testimonials or narratives from individuals who have recovered from COVID-19, emphasising their experience with the illness to highlight the continued relevance of vaccination as a protective measure for those who may perceive the risk as low or abstract.

Being a student offering a technical or applied course was significant in Model 2, but lost significance in Model 3. Studies have shown mixed patterns regarding programme of study, with some reporting higher acceptance among medical students [42,44,45] and others showing no difference [46] or higher acceptance in non-health fields [18]. The disappearance of significance in Model 3 suggests that the effect programme of study could be mediated by the other psychological constructs considered all together. For example, more than half of the students who reported ever having contracted COVID-19 were enrolled in the technical or applied programmes. It is plausible that their own illness or witnessing the experiences of classmates who had the disease may have influenced their perceptions and beliefs about vaccination, contributing to the observed significance in Model 2. Once constructs such as perceived susceptibility, perceived severity, attitude and self-efficacy were included in Model 3, the explanatory power of the programme of study may have been subsumed by these psychological variables.

Further, adherence to social distancing was also significantly associated with a higher intention to vaccinate. In the U.S. Northeast, clustering of vaccine-related attitudes and precautionary behaviours within social networks was reported [47]. That is, social network contacts following social distancing guidelines was significantly associated with the participant's intention to vaccinate [47]. Individuals who regularly adhere to social distancing measures likely share an underlying predisposition towards health-protective behaviours and a higher degree of perceived risk or collective responsibility [44]. Their willingness to follow one public health guideline (social distancing) could translate into a similar willingness to adhere to another (vaccination) which suggests a consistent health-conscious mindset.

Among TPB constructs, attitude emerged as a significant predictor of vaccination intention, consistent with evidence from a systematic review and meta-analysis [28]. Other studies have linked positive attitude to vaccination intention [39,43,44]. According to the TPB, the attitude of an individual towards a behaviour is a direct determinant of their intention to perform that behaviour [27–29]. A positive attitude reflects a belief that vaccination is desirable or beneficial [29]. Public health campaigns should strategically focus on shaping positive vaccination attitudes by emphasising the direct benefits of vaccination, such as protection from severe illness, hospitalisation, and death, and addressing common concerns about vaccine safety and efficacy. Public health campaigns should focus on shaping positive attitudes toward COVID-19 vaccination by reinforcing its perceived benefits such as protection from severe illness, and improved well-being among students. Health promotion activities should also directly address misinformation and misconceptions, including religious or apocalyptic beliefs and fears of long-term harm, while clarifying the safety, effectiveness, and ease of accessing the vaccine.

Subjective norms were a significant predictor of intention to vaccinate. A meta-analysis on TPB showed subjective norms had a strong effect size on vaccination intention [28]. Also, a Bangladeshi study confirmed that familial support (a component of subjective norms) significantly reduced vaccine hesitancy [43]. Furthermore, studies in the US highlighted that recommendations from family and friends were important mediators explaining demographic differences in vaccination uptake, and parental encouragement was a strong predictor of uptake among college students [48]. A national survey in Ghana also noted that encouragement by trusted community leaders increased vaccination intention [36]. Recommendations from close contacts and trusted figures significantly affect these intentions [49–51]. For students, social influence from peers and family can be particularly potent. If they perceive that their social circle or trusted figures approve of vaccination, they are more likely to intend to get vaccinated. Interventions could strengthen these influences by engaging peer educators, encouraging vaccinated students to share positive experiences, and involving respected community or institutional leaders as "vaccine champions".

In this study, except for self-efficacy, none of the other HBM constructs (perceived susceptibility, perceived severity, perceived benefits, perceived barriers, cues to action) retained statistical significance in the final model (Model 3) which contrasts a systematic review and meta-analysis study [40]. Studies have consistently identified it as a positive predictor of vaccination intention [29,49]. High self-efficacy is crucial for adopting protective behaviours and can help individuals overcome perceived barriers. Some studies using single-item measures for self-efficacy reported a lesser impact [48]. It implies that students who believe they can receive the COVID-19 vaccine on time, do not find it difficult to access vaccination services. These perceptions reflect their self-efficacy in managing the vaccination process. This confidence helps them overcome potential perceived barriers and translates directly into a stronger intention. Interventions could therefore include practical support measures such as streamlined access to vaccination points, on-campus vaccination days, clear instructions about procedures, and communication that reinforces vaccination as complementary to existing precautions.

Grounding this study on both HBM and TPB was able to explain about a quarter (25.7%) of the variance which is considerably higher compared to using the HBM (19%) and TPB (19.8%) separately. This explanatory power is somewhat lower than what has been reported in other Ghanaian studies, where similar behavioural models explained between 35% and 62.5% of the variance [52–54]. An Israeli study reported a significantly higher variance (78%) using same unified model [29]. One of possible explanation could be linked to the timing of vaccine availability. For instance, as the Israeli study was conducted in mid-2020 before vaccines were available, the Ghanaian studies and the present study were conducted after vaccine deployment which capture more practical, experience-based intentions [29,52–54]. Other reasons could be population characteristics, prior exposure to COVID-19, and differences in real-world vaccine experience [29,55].

## Limitations

This study has limitations that need to be considered while interpreting the findings. The cross-sectional design limits causal inference, so associations between predictors and vaccination intention cannot be interpreted as causal. All data were self-reported, introducing risks of social desirability and recall bias. In addition, findings are limited in generalisability, as the study was conducted in a single municipality among SHS students only. Internal consistency of some constructs also fell below accepted reliability thresholds, which may have weakened observed associations.

## Conclusions

Overall, about six in ten students expressed intention to vaccinate against COVID-19. This intention was influenced by prior experience with COVID-19, adherence to social distancing measures, positive vaccination attitudes, perceived social expectations, and confidence in one's ability to get vaccinated. Public health campaigns could involve students with personal COVID-19 experience as advocates and design strategies targeting attitudes, social norms, and confidence in accessing vaccination. Future interventions should strengthen students' self-efficacy and leverage social influence through peer and teacher engagement to enhance vaccine uptake in school settings.

## Supporting information

**S1 Data. A minimal dataset necessary to support the findings presented in the manuscript.**
(XLSX)

**S1 Table. Univariate analysis of COVID-19 vaccination intention by sociodemographic, health-related characteristics, and adherence to preventive protocols.**
(DOCX)

**S2 Table. Model fit statistics for hierarchical binary logistic regression models predicting intention to vaccinate against COVID-19.**
(DOCX)

**S3 Table. A summary of variable descriptions, value labels, and categorical classifications as used in the Stata analysis of the dataset supporting the findings presented in the manuscript.**
(DOCX)

## Acknowledgments

This research was conducted as part of Hudatu Ahmed's Bachelor of Public Health (Nutrition) degree at the Fred N. Binka School of Public Health, University of Health and Allied Sciences, Ghana. The authors acknowledge the Education Director, headteachers, and teachers of the Jasikan Municipality, Ghana for their support in facilitating data collection. Appreciation is also extended to the trained research assistants and health staff for their role in data collection, and to all study participants for their participation.

## Author contributions

**Conceptualization:** Hudatu Ahmed, Mawuli Gohoho, Veronica Okwuchi Charles-Unadike.

**Data curation:** Hudatu Ahmed, Mawuli Gohoho, Samuel Adolf Bosoka.

**Formal analysis:** Hudatu Ahmed, Mawuli Gohoho, Samuel Adolf Bosoka, Sorengmen Amos Ziema, Veronica Okwuchi Charles-Unadike.

**Investigation:** Hudatu Ahmed, Mawuli Gohoho, Veronica Okwuchi Charles-Unadike.

**Methodology:** Hudatu Ahmed, Mawuli Gohoho, Samuel Adolf Bosoka, Veronica Okwuchi Charles-Unadike.

**Visualization:** Hudatu Ahmed, Mawuli Gohoho, George Sarpong Agyemang, Sorengmen Amos Ziema, James Alorwu, Veronica Okwuchi Charles-Unadike.

**Writing – original draft:** Hudatu Ahmed, Mawuli Gohoho.

**Writing – review & editing:** Hudatu Ahmed, Mawuli Gohoho, Samuel Adolf Bosoka, George Sarpong Agyemang, Sorengmen Amos Ziema, James Alorwu, Isaac Annobil, Veronica Okwuchi Charles-Unadike.

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
