## [Decision Letter · Decision Letter 0]

11 Nov 2025

PGPH-D-25-03045

Predictors of COVID-19 vaccination intention using Health Belief Model and Theory of Planned Behaviour: A cross-sectional study in Jasikan Municipality, Ghana

Dear Dr. Mawuli Gohoho,

Thank you for submitting your manuscript to PLOS Global Public Health. After careful consideration, we feel that it has merit but does not fully meet PLOS Global Public Health’s publication criteria as it currently stands. Therefore, we invite you to submit a revised version of the manuscript that addresses the points raised during the review process.

We look forward to receiving your revised manuscript.

Kind regards,

Titilayo Abike Olaoye, PhD

Academic Editor

Journal Requirements:

1. Your manuscript is missing the following sections: Abstract. Please ensure these are present, and in the correct order, and that any references to subheadings in your main text are correct. An outline of the required sections can be consulted in our submission guidelines here: 

https://journals.plos.org/globalpublichealth/s/submission-guidelines#loc-parts-of-a-submission

2. Please provide separate figure files in .tif or .eps format.

https://journals.plos.org/climate/s/figures 

https://journals.plos.org/climate/s/figures#loc-file-requirements 

Additional Editor Comments:

This is a valuable and timely paper that explores behavioural predictors of COVID-19 vaccination among students in Ghana, using the HBM and TPB frameworks. The topic aligns well with PLOS Global Public Health’s focus on health equity and behavioural determinants. However, substantial revision is required to improve the manuscript’s clarity, methodological rigor, conceptual depth, and adherence to journal standards.

Section-by-Section Comments

Title and Abstract

Lines 1–15: Title is clear but overly long. Suggest simplifying to:

• “Predictors of COVID-19 vaccination intention among students in Ghana: An application of the Health Belief Model and Theory of Planned Behaviour.”

• Abstract (Lines 18–51): The background section repeats detailed context (e.g., “in 2021 most COVID-19 cases…”), which should be shortened.

• Methods lack clarity about how HBM and TPB constructs were operationalised—specify how scales were developed.

• Conclusion is appropriate but should explicitly link findings to potential interventions.

Introduction (Lines 54–137)

• Lines 54–61: The first paragraph reads as a general background on vaccines; it can be shortened to 3–4 sentences.

• Lines 62–91: Context on Ghana’s COVID-19 response is too detailed (dates, COVAX timeline, etc.). Retain only what informs local vaccine hesitancy.

• Lines 92–137: The justification for using both HBM and TPB is good but poorly synthesized—needs a clear conceptual rationale for integrating the two frameworks rather than describing them separately.

• End the section with explicit research objectives or hypotheses (currently missing).

• Consider adding a conceptual diagram early in the Introduction showing the link between HBM, TPB, and intention.

Methods (Lines 139–270)

• Study design (Lines 162–171): Clearly state why a cross-sectional design was chosen and its limitations for causal inference.

• Sample size (Lines 172–185):

• The formula is presented, but the expression is broken and not formatted properly.

• Add information on response rate and handling of non-response.

• Data collection (Lines 186–197):

• Clarify whether data collectors were blinded to study hypotheses.

• Describe translation, pretesting process, and reliability assessment for all items.

• Variables (Lines 198–231):

• Many constructs are listed but not referenced—cite validated scales or justify modifications.

• Dichotomisation of intention may lead to information loss; discuss this decision.

Statistical analysis (Lines 250–270):

Explain criteria for including variables in multivariable models (e.g., p<0.2 from univariate?).

State how multicollinearity was checked.

Clarify whether model assumptions (e.g., goodness of fit) were tested.

Add description of missing data handling.

Results (Lines 271–379)

• Tables: Well-structured but lengthy. Some could be moved to supplementary materials (e.g., full univariate table).

• Lines 318–343: Hierarchical models are described clearly, but the rationale for three-step modeling should be briefly restated.

• Statistical interpretation:

• Avoid redundant p-values when AORs and CIs are reported.

• Emphasise magnitude and direction of effects, not just significance.

• Figures (e.g., Fig. 2) should be properly labeled with titles and legends conforming to PLOS format.

Discussion (Lines 381–533)

• Lines 381–414: The first paragraph is too long; break it into subtopics (comparison with other studies, interpretation of findings).

• Lines 415–433: Interpretation of “history of COVID-19 infection” as a predictor is good but too descriptive—consider theoretical explanation via “perceived susceptibility” and “cue to action.”

• Lines 433–479: Several comparisons with foreign studies are excessive. Summarise key consistencies and differences in one paragraph.

• Lines 480–504: The interpretation of self-efficacy and subjective norms is strong—expand on implications for intervention design.

• Lines 511–533: The discussion of model explanatory power (R² = 25.7%) is informative but needs statistical context—consider referencing similar behavioral studies in Africa.

• Limitations (Lines 535–569): Comprehensive but verbose. Condense to five concise sentences, prioritising: cross-sectional design, self-report bias, limited generalisability, and moderate internal reliability.

Conclusion (Lines 571–579): Currently reiterates findings; should include specific actionable recommendations for policymakers or school-based vaccination programmes. For instance, “Future interventions should strengthen students’ self-efficacy and leverage social influence through peer and teacher engagement to enhance vaccine uptake in educational settings.”

Technical and Formatting Issues

• Referencing: Inconsistent citation style; some references appear as numeric, others as author–year in-text (e.g., [35,36]). Conform to PLOS Vancouver format.

• Language and grammar: Frequent redundancy, long sentences, and inconsistent tense usage. Recommend full professional language editing.

• Tables/Figures: Ensure all are cited in order and formatted per journal templates.

• Data availability: Statement is acceptable but should include a repository link (e.g., OSF, Dryad) as per PLOS policy.

Reviewers' comments:

Reviewer's Responses to Questions

**Comments to the Author**

1. Does this manuscript meet PLOS Global Public Health’s publication criteria?

Reviewer #1: Yes

2. Has the statistical analysis been performed appropriately and rigorously?

Reviewer #1: Yes

3. Have the authors made all data underlying the findings in their manuscript fully available (please refer to the Data Availability Statement at the start of the manuscript PDF file)?

Reviewer #1: Yes

4. Is the manuscript presented in an intelligible fashion and written in standard English?

Reviewer #1: Yes

Reviewer #1: The authors of the manuscript titled 'Predictors of COVID-19 vaccination intention using Health Belief Model and Theory of Planned Behaviour: A cross-sectional study in Jasikan Municipality, Ghana" has contributed to knowledge especially in the area of literature review on vaccination intention. I have commented on the few lines the authors will have to make adjustments.

**Do you want your identity to be public for this peer review?** For information about this choice, including consent withdrawal, please see our Privacy Policy

Reviewer #1: **Yes: ** Oladunni Opeyemi

---

## [Editor Report · Decision Letter 1]

9 Dec 2025

Predictors of COVID-19 vaccination intention among students in Ghana: An application of the Health Belief Model and Theory of Planned Behaviour

PGPH-D-25-03045R1

Dear Dr Mawuli Gohoho

We are pleased to inform you that your manuscript 'Predictors of COVID-19 vaccination intention among students in Ghana: An application of the Health Belief Model and Theory of Planned Behaviour' has been provisionally accepted for publication in PLOS Global Public Health.

Best regards,

Titilayo Abike Olaoye, PhD

Academic Editor